# Parallel Cellular Automata Markov Model for Land Use Change Prediction over MapReduce Framework

**Junfeng Kang** [1,2] , **Lei Fang** [3,]*, **Shuang Li** [1] **and Xiangrong Wang** [3]

[1]  School of Earth Sciences, Zhejiang University, Hangzhou 310027, China; junfeng.kang@zju.edu.cn (J.K.); namespacezhang@mail.jxust.edu.cn (S.L.)

[2]  School of Architectural and Surverying & Mapping Engineering, Jiangxi University of Science and Technology, Ganzhou 341000, China

[3]  Department of Environmental Science and Engineering, Fudan University, Shanghai 200438, China; xrxrwang@fudan.edu.cn

*  Correspondence: fanglei@fudan.edu.cn

**Abstract:** The Cellular Automata Markov model combines the cellular automata (CA) model's ability to simulate the spatial variation of complex systems and the long-term prediction of the Markov model. In this research, we designed a parallel CA-Markov model based on the MapReduce framework. The model was divided into two main parts: A parallel Markov model based on MapReduce (Cloud-Markov), and comprehensive evaluation method of land-use changes based on cellular automata and MapReduce (Cloud-CELUC). Choosing Hangzhou as the study area and using Landsat remote-sensing images from 2006 and 2013 as the experiment data, we conducted three experiments to evaluate the parallel CA-Markov model on the Hadoop environment. Efficiency evaluations were conducted to compare Cloud-Markov and Cloud-CELUC with different numbers of data. The results showed that the accelerated ratios of Cloud-Markov and Cloud-CELUC were 3.43 and 1.86, respectively, compared with their serial algorithms. The validity test of the prediction algorithm was performed using the parallel CA-Markov model to simulate land-use changes in Hangzhou in 2013 and to analyze the relationship between the simulation results and the interpretation results of the remote-sensing images. The Kappa coefficients of construction land, natural-reserve land, and agricultural land were 0.86, 0.68, and 0.66, respectively, which demonstrates the validity of the parallel model. Hangzhou land-use changes in 2020 were predicted and analyzed. The results show that the central area of construction land is rapidly increasing due to a developed transportation system and is mainly transferred from agricultural land.

**Keywords:** CA Markov; land-use change prediction; Hadoop; MapReduce; cloud computing

## 1. Introduction

Studying land use/land cover changes in different times and places and predicting land-use structures and spatial layouts can provide scientific support for the utilization of regional land resources, the protection of regional ecological environments, and sustainable social and economic development [1,2].

Many researchers have proposed their own land-use-change simulation and prediction models, such as CLUE [3], CLUE-S [4], cellular automata (CA) [2,5,6], Markov chain [7,8], SLEUTH [9,10], and the spatial logistic model [11]. Since the early 1980s when Wolfram first proposed the CA model [12], many studies have been conducted to use the CA model to simulate urban land-use changes [2,5,6,13] and researchers have integrated other methods or models, such as neural networks [14], support vector

machines (SVM) [15], ant-colony optimization [16], and the Markov chain [17], into the CA model to simulate and monitor land-use change.

The CA-Markov model was one of the most widely used extended CA models and was used in the prediction and simulation of land-use changes in many countries, such as the United States [18], Brazil [19], Portugal [20], Egypt [21], Ethiopia [22], Bangladesh [23], Malaysia [24], and China [25]. It has also been applied to the research of the evolution of urban-settlement patterns [26], the process of spatial dynamic vegetation changes [27], and land transfer across metropolitan areas [28].

As land-use change simulation and prediction involves tremendous numbers of data and calculations, in recent years, some studies have designed parallel CA algorithms on Central Processing Unit (CPU) parallel computing [29], Message Passing Interface (MPI) [30], Graphics Processing Unit (GPU) parallel [31], and GPU/CPU hybrid parallel [32] to simulate urban growth. However, the parallel CA method cannot deal with the connection between partitions after a research area is divided into several pieces, resulting in different final-prediction results, whereas the traditional Markov method is able to maintain the integrity of the entire study area but results in a lack of spatial relationship of the land cell. On the other hand, with the development of Big Data technologies and applications, MapReduce is a promising method to improve traditional-serial-algorithm running efficiency and has been applied and proven effective in many cases. Rathore et al. [33] proposed a real-time remote-sensing image processing and analysis application. Raojun et al. [34] proposed a parallel-link prediction algorithm based on MapReduce. Wiley K. et al. [35] analyzed astronomical graphics based on MapReduce, while Almeer [36] used Hadoop to analyze remote-sensing images, improving batch-reading and -writing efficiency. For the CA Markov model, the MapReduce framework is not only capable of efficient parallel processing, but can also be a coupling to the CA-Markov model: The "Map" corresponds to the CA process to realize the parallelism of land-use-unit-change prediction; "Reduce" refers to the Markov process to achieve overall prediction of land-use changes. However, because the key problem of segmentation and connection remains unresolved, there is little research on the parallel CA-Markov model for land-use-change prediction over the MapReduce framework.

Based on in-depth analysis of the parallelism of the CA Markov model, this paper first proposes a parallel solution that uses the MapReduce framework to improve the CA Markov model for land-use-change prediction. The parallel CA-Markov model can not only solve the contradiction that the traditional CA-Markov model cannot simultaneously realize the integrity and segmentation for land-use change simulation and prediction, but can also ensure both efficiency and accuracy and realize land-use change prediction in a cloud-computing environment.

## 2. Materials and Background Technologies

### 2.1. Study Area and Data

Hangzhou is located in the southeast coast of China, which is the political, economic, cultural, and financial center of Zhejiang Province. Hangzhou has a complex topography: The west is a hilly area with the main mountain range, including Tianmu Mountain, and the east is a plain area with low-lying terrain and dense river networks.

The 2006 Landsat TM and the 2013 Landsat8 remote-sensing image with 30 m resolution of the study area were downloaded from http://www.gscloud.cn/. Other experiment datasets included DEM with a 30 m resolution, road-network data, traffic-site data, and location-address data.

Nowadays, many researches develop their own auto image-identification method to classify the high-resolution image, especially unmanned aerial vehicle (UAV) images [37,38]. Because Landsat images were the medium-resolution image, we chose to use a semi-manual method to preprocessed and interpreted to obtain land-use data using ENVI 5.3 and ArcGIS 10.2. [39]. As shown in Figure 1, the workflow of Landsat images classification included four main steps, namely definition of the classification inputs, preprocessing, region of interest, and classification. The processing steps mainly included geocorrection, geometric rectification, or image registration, radiometric calibration and

atmospheric correction, and topographic correction [40]. A support-vector-machine (SVM) classifier was selected for land-use classification [41].

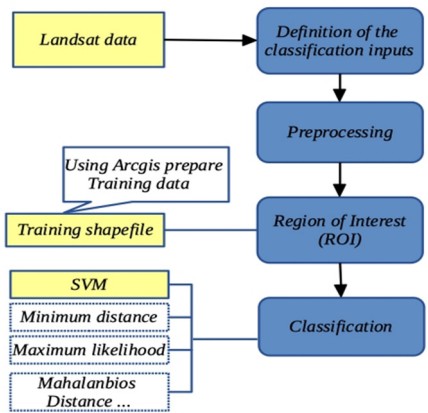

**Figure 1.** Landsat-image classification flow.

In general, land-use types include cultivated land, forest, grassland, waters, construction land, gardens, transportation land, unused land, and swampland. In our experiment, four land-use types were defined in the training shapefile after reclassifying. The land-use-type reclassification method of construction land (B), agricultural land (A), nature reserve (N), and water area (W) are defined as shown in Table 1.

**Table 1.** Land-use reclassifications.

| Level 1. | Level 2 | Definition |
|---|---|---|
| Construction land (B) | Land for construction (B1), land for transportation (B2) | Land for buildings and structures. |
| Agricultural land (A) | Cultivated land (A1), garden (A2) | Land for agricultural production. |
| Water area (W) | Waters (W1), swampland (W2) | River surface, lake surface, swamp. |
| Nature reserve (N) | Forest (N1), grassland (N2), unused land (N3) | Land with little or no human activity that did not include agricultural land, construction land, and waters. |

### 2.2. MapReduce

The MapReduce [42] program consists of two functions, Map and Reduce. Both of these functions take key/value (key-value pair) as input and output. The Map function receives a user-entered key-value pair $(k_1, v_1)$ and processes it to generate a key-value pair $(k_2, v_2)$ as an intermediate result. Then, the corresponding values of all the same intermediate keys $(k_2)$ are aggregated to generate a list of values for the $k_2$ key list $(v_2)$, which is used as an input to the Reduce function and processed by the Reduce function to obtain the final result list $(k_3, v_3)$. The process can be expressed by the following formula:

$$\text{Map} : (k_1, v_1) - \text{list}(k_2, v_2) \tag{1}$$

$$\text{Reduce} : (k_2, \text{list}(v_2)) - \text{list}(k_3, v_3) \tag{2}$$

The MapReduce framework is shown in Figure 2:

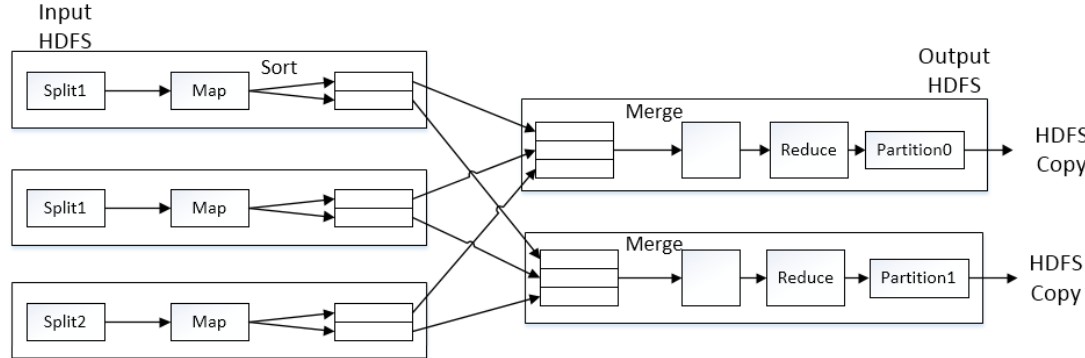

**Figure 2.** Overview of MapReduce process framework.

### 2.3. CA Markov Model

The Markov model is based on the theory of random processes. In this model, given the initial state and state-transition probabilities, simulation results have nothing to do with the historical condition before the current condition, which can be used to describe land-use changes from one period to another. We can also use this as a basis to predict future changes. Change was found by creating a land-use-change transition-probability matrix from periods $t$ to $t + 1$, which is the basis to predict future land-use changes [43].

$$S(t+1) = P_{ij} \times S(t) \tag{3}$$

$S(t + 1)$ denotes the state of the land-use system at times $t + 1$ and $t$, respectively. $P_{ij}$ is the state-transition matrix.

CA has four basic components: Cell and cell space, cell state, neighborhood, and transition rules. The CA model can be expressed as follows [44]:

$$S(t+1) = f(S(t), N) \tag{4}$$

In the formula, $S$ is a state set of a finite and discrete state, $t$ and $t + 1$ are different moments, N is the neighborhood of the cell, and f is the cell-transition rule of the local space.

Usually, in order to make cellular automata better simulate a real environment, space constraint variable $\beta$ needs to be introduced to express the topographic terrain, as well as the adaptive constraints and restrictive constraints of the spatial-influence factors on the cells. The formula becomes

$$S(t+1) = f(S(t), N, \beta). \tag{5}$$

The separate Markov model lacks spatial knowledge and does not consider the spatial distribution of geographic factors and land-use types, while the CA-Markov model adds spatial features to the Markov model, uses a cellular-automata filter to create weight factors with spatial character, and changes the state of the cells according to the state of adjacent cells and the transition rules [21].

## 3. Methods

### 3.1. Parallel CA-Markov Model Overview

#### 3.1.1. Parallel CA-Markov Structure

Figure 3 is the structure of parallel CA-Markov over MapReduce framework. The structure contains four layers from bottom to top: The data layer, parallel CA-Markov model layer, land-use transition-direction determination layer, and land-use prediction layer.

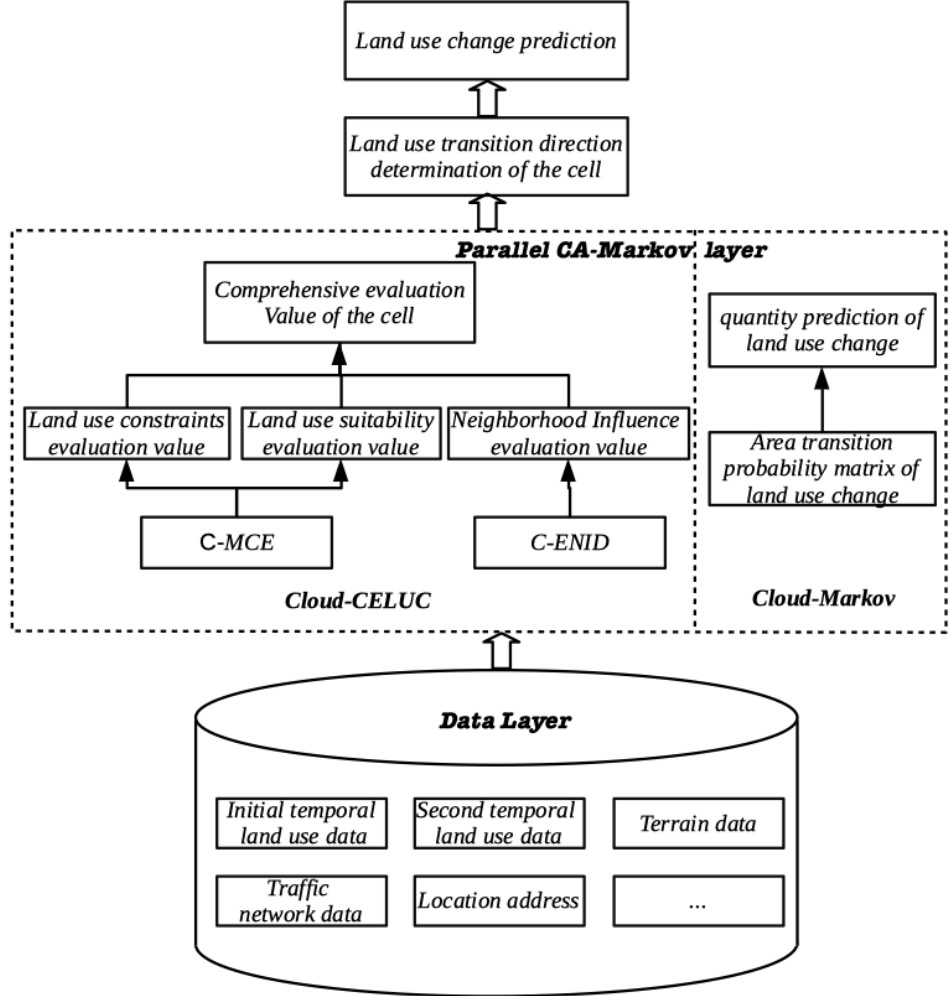

**Figure 3.** Structure of parallel cellular-automata Markov over MapReduce.

The experiment data include two-phase remote-sensing images, terrain data, traffic-network data, and location-address data.

The parallel CA-Markov model can mainly be divided into two parts: A parallel Markov algorithm based on MapReduce (Cloud-Markov), and a comprehensive evaluation method of land-use changes based on MapReduce (Cloud-CELUC).

Cloud-Markov was used to calculate the area transition-probability matrix of land-use changes. The algorithm of Cloud-Markov is discussed in Section 3.2.

Cloud-CELUC includes three main parts: Evaluation of the influence of neighborhoods under a cloud-computing environment (C-ENID), multicriteria evaluation under a cloud-computing environment (C-MCE), and comprehensive evaluation of land use. Among them, C-ENID is a parallel CA model over MapReduce, and C-MCE is a MapReduce algorithm to calculate constrained-evaluation and suitability-evaluation values. The cell's neighborhood influences and C-ENID's calculation method is discussed in Section 3.3.1. C-MCE design factors are discussed in Section 3.3.2, and the Cloud-CELUC algorithm is discussed in detail in Section 3.3.3.

After using Cloud-Markov to calculate the transition-probability matrix of land-use changes and using Cloud-CELUC to calculate each cell's comprehensive evaluation value, each cell's land-use transition direction was determined, which is discussed in Section 3.4. Finally, land-use-change predictions were obtained, and the land-use prediction experiment is discussed in Section 4.

### 3.1.2. Parallel CA-Markov Workflow

As shown in Figure 4, the flow of the parallel CA-Markov model has five major steps, as follows:

(1) Data-processing: Preprocessing and interpreting remote-sensing images to land-use maps, designing multicriteria-evaluation factors, and storing images, land-use data, and multicriteria-evaluation factors into the Hadoop HDFS.

(2) Parallel Markov: Using the overlay method, analyzing two-phase images and land-use data to obtain each cell's land-use-type transition probability, and calculating the total number of cells in each land-use-type transition direction, and counting the area transition-probability matrix of each land-use type.

(3) Parallel CA: In Cloud-CELUC, C-ENID was used to calculate the cell's neighborhood influence value. C-MCE was designed to calculate multicriteria-evaluation values, including constraint-evaluation and suitability-evaluation values. These values were then used to calculate the statistical table of comprehensive evaluation of land use.

(4) Transition-direction determination stage: Loop reading each cell's transition probability from the statistical table of comprehensive-evaluation values in the parallel CA stage and combining the area transfer-probability matrix of each land-use type in the parallel Markov stage to decide a cell's land-use-type transition direction.

(5) Land-use-change prediction: In our experiment, we used data from 2006 to predict 2013 land-use changes and then evaluate the precision of the parallel CA-Markov model with a Kappa coefficient. Land-use change prediction for 2020 was then obtained.

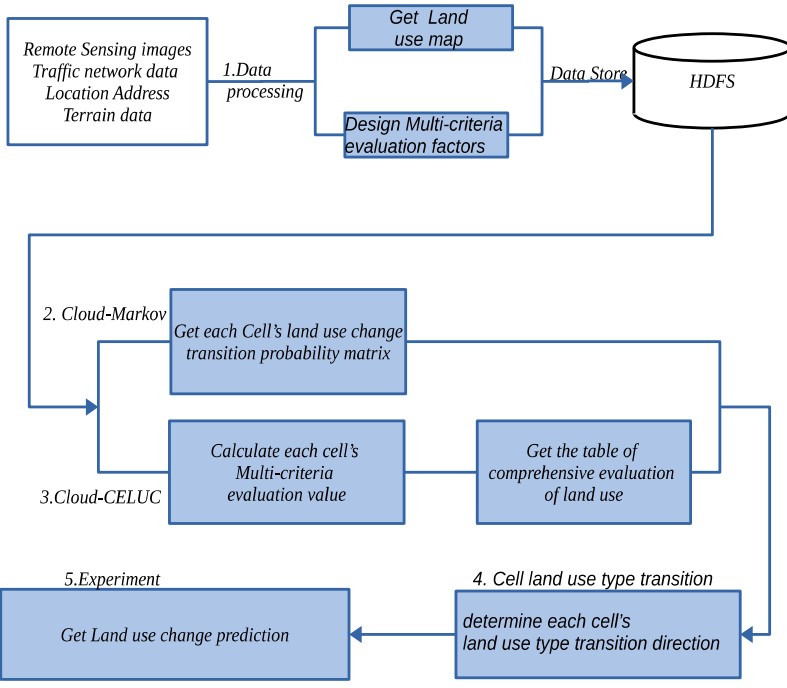

**Figure 4.** Parallel Cellular Automata-Markov (CA-Markov) flow.

### 3.2. *Markov Model Parallel Processing (Cloud-Markov)*

Using an overlay method to analyze two-phase raster images and land-use data with the same spatial position, we obtained the transition direction of each cell and calculated the number of cells in each transition direction, then obtained the area transfer matrix of each land-use type and calculated the area probability matrix of land use. Each year's area transfer matrix was then obtained using the

probability matrix divided by the intervals of these two raster images. Then, MapReduce functions of the Markov model given in the study area are as follows:

$$\text{Map} : (N, (T_1, T_2)) \rightarrow List\left(C_{mk}, i\right) \tag{6}$$

$$\text{Combiner} : M\left(C_{mk}, list(i)\right) \rightarrow List\left(C_{mk}, \text{s}\right) \tag{7}$$

$$\text{Reduce} : L\left(C_{mk}, list(i)\right) \rightarrow List\left(C_{mk}, \text{s}\right) \tag{8}$$

where $N$ is the row offset of the input row, $T_1$ represents all cells' land-use type of the earlier raster image, $T_2$ represents all cells' land-use type of the later raster image, $C_{mk}$ represents the cell conversion from land-use type $m$ to land-use type $k$, $i$ indicates the cell transfer number of $C_{mk}$, $M$ is the number of land-use types in a MapReduce node, $s$ is the total cell number of $C_{mk}$ in a MapReduce node, $L$ is the total number of land-use types, and $q$ is the combined value of the total number of cells and transition probability for $C_{mk}$ conversion in the entire study area. For example, "100-3.12%" means that there were 100 cells with $C_{mk}$ conversion, and transition probability was 3.12%. Figure 5 is the flow of Cloud-Markov algorithm. After summing the number of same land-use type transition direction in each node, the area transition matrix was calculated at the Reduce stage.

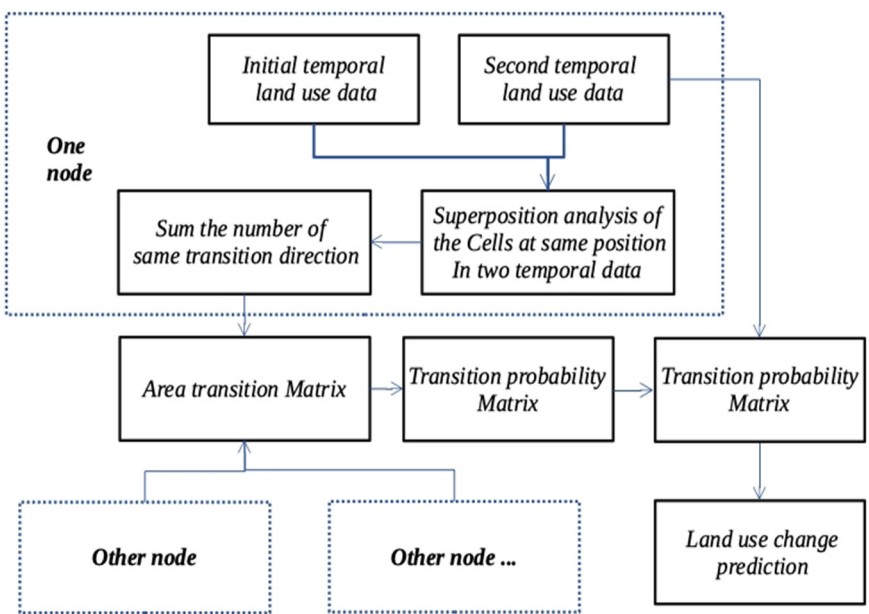

**Figure 5.** Cloud-Markov algorithm flow.

The steps are as follows:

(1)    Map stage:

    ①    Input <Key,Value>
    ②    Raster cell's land-use type conversion analysis

In this step, by comparing the cells in the same position between these two raster images, we obtained a list of $C_{mk}$. If the value of $C_{mk}$ is 'B-A', it means that the land-use-type of the cell with the same position in different raster images is converted from B into A.

    ③    Output <Key,Value>

where key is conversion direction $C_{mk}$, and Value is an integer equal to 1.

(2)    Combiner stage:

　① 　Enter <Key,Value>
　② 　Calculate the number of each land-use-type conversion direction in each node

<Key,Value> is a key-value pair ($C_{mk}$,$s$), where $C_{mk}$ is the conversion direction and $s$ is the total number of cells in a MapReduce node where the $C_{mk}$ land-use-type conversion direction occurs.

　③ 　Output <Key,Value>

Output key-value pairs ($C_{mk}$,$s$).

(3) Reduce stage:

　① 　Enter <Key,Value>
　② 　Calculate transition probability

The transition-probability-matrix calculation formula is defined as follows:

$$P_{mk} = \frac{V_{mk}}{S_m} \tag{9}$$

where $V_{mk}$ is the value that summed $C_{mk}$ from all MapReduce nodes and $S_m$ is the number of the initial raster image's cells whose land-use type is $m$.

　③ 　Output <Key,Value>

<Key,Value> is a key-value pair ($V_{mk}$,$P_{mk}$), where $V_{mk}$ is the land-use-type area-conversion matrix and $P_{mk}$ is the transition-probability matrix.

*3.3. Cloud-CELUC*

3.3.1. Cell Neighborhood Processing

Obtaining a cell's neighborhood-influence value requires reading the state of the neighbor cells. The general neighborhood-influence evaluation methods include Von Neumann and Moore. Figure 6 shows how to read cellular-neighborhood, where we chose the $3 \times 3$ Moore method to design our algorithm.

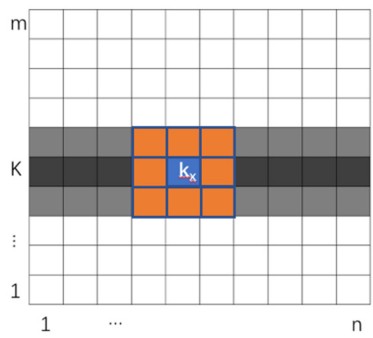

**Figure 6.** Cellular-neighborhood reading.

Figure 7 shows the process of cellular-dimension reduction, where a list structure was used to store all cells, so we could read each cell's neighborhood cells through the cell's row index and column index. The two-dimensional raster image was reduced into a one-dimensional array that could reduce the data exchange between each node during MapReduce processes. For example, the neighborhood cells of cell $K_x$ were from line $K-1$, $K$, and $K+1$, were recorded as $K-1_{x-1}$, $K-1_x$, $K-1_{x+1}$, $K_{x-1}$, $K_{x+1}$, $K+1_{x-1}$, $K+1_x$, and $K+1_{x+1}$, and then stored as an array structure into the HDFS.

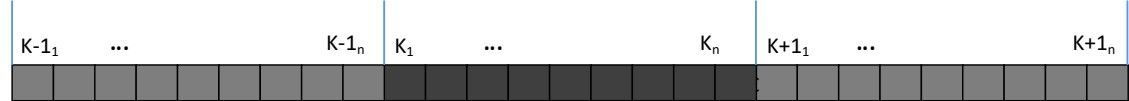

**Figure 7.** Cellular-dimension reduction.

### 3.3.2. Multicriteria Evaluation Factors

This research used the multicriteria evaluation (MCE) method to calculate constraint-evaluation and suitability-evaluation values. The purpose of MCE is to select the optimal decision solution in limited (infinite) solutions between which there are conflicts and coexistence [45].

The evaluation criteria of MCE are divided into two parts: Suitable factors and constraint factors. Suitable factors are used to normalize the influencing factor to the continuous measurable values, and constraint factors are utilized to categorize spatial features by their spatial characteristics. The value of the factors is a Boolean type with a value of 0 or 1.

Suitability factors are defined and standardized in Table 2, where eight factors are defined to distinguish the distance from a cell to different typical destination, and according to different land-use type, the weighted value of each factor is defined in Table 3. Then, the analytic hierarchy process (AHP) method and the expert scoring method were used to calculate the weights of the suitability factors [46], which are shown in Table 2. The waters and gradient factors were defined as the constraint factors.

**Table 2.** Suitability-factor classification.

| Factor Name | Definition | Classification |
| --- | --- | --- |
| FreLev<br>TownLev | Distance from cell to highway.<br>Distance from cell to town center. | Cell distance from main road or town center: 0–250, 250–500, 500–750, 750–1000, and 1000–1250 m. |
| SubLev<br>BusLev<br>MainLev | Distance from cell to subway station.<br>Distance from cell to bus stop.<br>Distance from cell to other roads. | Cell distance to subway or bus station, other roads: 0–100, 100–200, 200–300, 300–400, and 400–500 m. |
| TraLev<br>StaLev | Distance from cell to train station.<br>Distance from cell to bus station. | Cell distance to train or bus station: 0–200, 200–400, 400–600, 600–800, and 800–1000 m. |
| CityLev | Distance from cell to county center. | Cell distance to main road or county center: 500–1000, 1000–1500, 1500–2000, and 2000–2500 m. |

**Table 3.** Weight parameters of suitability factors.

| Factor Name | Agricultural Land | Construction Land | Nature Reserve |
| --- | --- | --- | --- |
| FreLev | 0.0485 | 0.0461 | 0.0781 |
| TownLev | 0.1239 | 0.1332 | 0.1010 |
| SubLev | 0.0621 | 0.1320 | 0.0133 |
| BusLev | 0.0721 | 0.1110 | 0.0513 |
| MainLev | 0.0921 | 0.1102 | 0.0749 |
| TraLev | 0.0423 | 0.1333 | 0.0201 |
| StaLev | 0.0623 | 0.1321 | 0.0203 |
| StaLev | 0.0923 | 0.2021 | 0.0103 |

The value of the constraint factors is Boolean, which is determined by land-use type. This research defined the constraint factors of hills with gradients greater than 25 degrees, and waters and ecological reserves equal to 0, because these land-use types would rarely change.

### 3.3.3. Parallel Cloud-CELUC Algorithm

Cloud-CELUC only needs the Map function to calculate factors and obtain comprehensive-evaluation values. The Reduce function of Cloud-CELUC was only used to output the results. The Map function was defined as follows:

$$\text{Map} : (N, (i, H_1, H_2, H_3, H_4, \ldots H_m)) \rightarrow ((i, j), (L_1, L_2, L_3, \ldots L_m)) \tag{10}$$

where $N$ is the line offset of the input line, $i$ is the line index of the raster image, $j$ is the column index of the raster image, $H_1$ is the cell-state value to be calculated, and $H_2$ and $H_3$ indicate the uplink and downlink cell-state values of $H_1$. $H_4, \ldots H_m$ are various constraint factors and suitability factors corresponding to the cell, and $L_m$ is the value combined by the cell's composite-evaluation value of the transition direction and the cell's corresponding transition direction. For example, If $L_1$ is 'ba-1.2234', the evaluation value of the cell $(i, j)$ from the initial land-use type 'b' to the final land-use type 'a' is '1.2234'.

The flow of Cloud-CELUC algorithm is shown in Figure 8, where the table of comprehensive evaluation value was obtained after each node by calculating the neighborhood impact value, suitability evaluation value, and constrained evaluation value. The steps are as follows:

① Enter <Key,Value>

② Calculate neighborhood-influence evaluation value (*NID*)

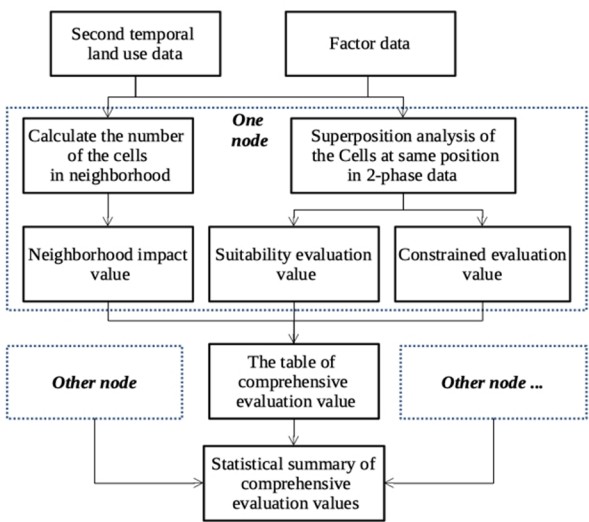

**Figure 8.** Cloud-comprehensive-evaluation value (CELUC) algorithm.

According to $H_1$ and $H_3$, the neighborhood-influence degree of each cell in $H_2$ was calculated. The calculation formula for the evaluation value of the neighborhood-influence degree of the cell $(i,j)$ corresponding to a class at a certain time is as follows:

$$NID_a = \frac{1}{h-1} \sum Yes\left(S_{ij} == a\right) \tag{11}$$

where $h$ is the number of neighborhood cells, and $Yes\left(S_{ij} == a\right)$ is used to judge whether the neighbor cell's land-use type of cell $(i, j)$ is $a$ or not. If $a$ is 1, then $Yes\left(S_{ij} == a\right)$ returns 1, otherwise, it returns 0.

③ Calculate Suitability-Evaluation Value (*SEV*)

The calculation formula of the suitability-evaluation value is as follows:

$$SEV_a = \sum V_{a\delta} \times DIS_{ij\delta} \tag{12}$$

where $V_{a\delta}$ is the weight of factor $\delta$ corresponding to land-use type '*a*', $DIS_{ij\delta}$ are the suitability factors of cell (*i,j*) that are defined in Table 1.

④     Calculate constraint-evaluation value (*CEV*)

The constraint-evaluation formula is as follows:

$$CEV_a = \prod Yes\left(K_{ij}\right) \tag{13}$$

where $Yes\left(K_{ij}\right)$ represents the constraint-evaluation value of cell (*i, j*) corresponding to constraint factor '*k*'. If the cell is constrained, $Yes\left(K_{ij}\right)$ returns 0, otherwise, it returns 1.

⑤     Calculate comprehensive-evaluation value (*CELUC*)

Based on the three evaluation values above, *NID, SEV,* and *CEV,* the comprehensive-evaluation formula was defined as follows:

$$CELUC_a = NID_a \times SEV_a \times CEV_a \tag{14}$$

where *a* is a land-use type.

⑥     Output <Key,Value>

<Key,Value> is a key-value pair ((*i, j*), *CELUC*) where *i* is the cell's row index, *j* is the cell's column index, and *CELUC* is the comprehensive-evaluation value of the cell.

### 3.4. Cell Land-Use-Type Conversion

The multi-objective land-use competition method was used to achieve cells' land-use-type conversion, which was to solve the problem when the cell had confliction in land-use-type conversion [47]. For example, if there were N types of land-use types, cell (*i, j*) may have had N kinds of conversion possibilities. According to constraint factors, suitability factors, and neighborhood conditions, each cell's conversion possibility should be given a comprehensive evolution value of land use. If the evaluation value of the cell's transition direction is determined by the biggest conversion possibility, the value may lead to a proliferation of dominant land-use types and cause oversimulation of dominant land-use types and inadequate simulation of weak land-use types. Hence, both land-use comprehensive evaluation values and the area transfer matrix of land-use changes were used to decide the cell's conversion direction. Figure 9 shows the flow of a cell's land-use-type conversion process.

The steps are as follows:

①     Calculating the maximum evaluation value from the table of statistical summary of comprehensive evaluation that was obtained from Cloud-CELUC.

②     Loop reading each row of the table. Each row was a key-value pair ((*i, j*), CELUCs), where (*i, j*) is the position of the cell, CELUCs means cell (*i, j*) has N kinds of land-use conversion possibilities (CELUC), and CELUC*i* means the *i*th CELUC of the cell.

③     Determining whether the area of the converted land-use type reached the upper area limit of this land-use-type conversion or not.

The upper area limit of the land-use-type conversion was obtained from the area transition-probability matrix of each land-use type, defined as a key-value pair ($V_{mk}$, $P_{mk}$) where $V_{mk}$ is the land-use-type area conversion matrix and $P_{mk}$ is the land-use transition-probability matrix at the CLOUD-Markov stage.

④     If reaching the upper area limit, the CELUC*i* of the cell should be marked as 0, meaning that one of the cell (*i, j*)'s CELUC*i* was deleted to make sure the CELUC*i* would not be used in the subsequent steps. Then, it returns to the first step.

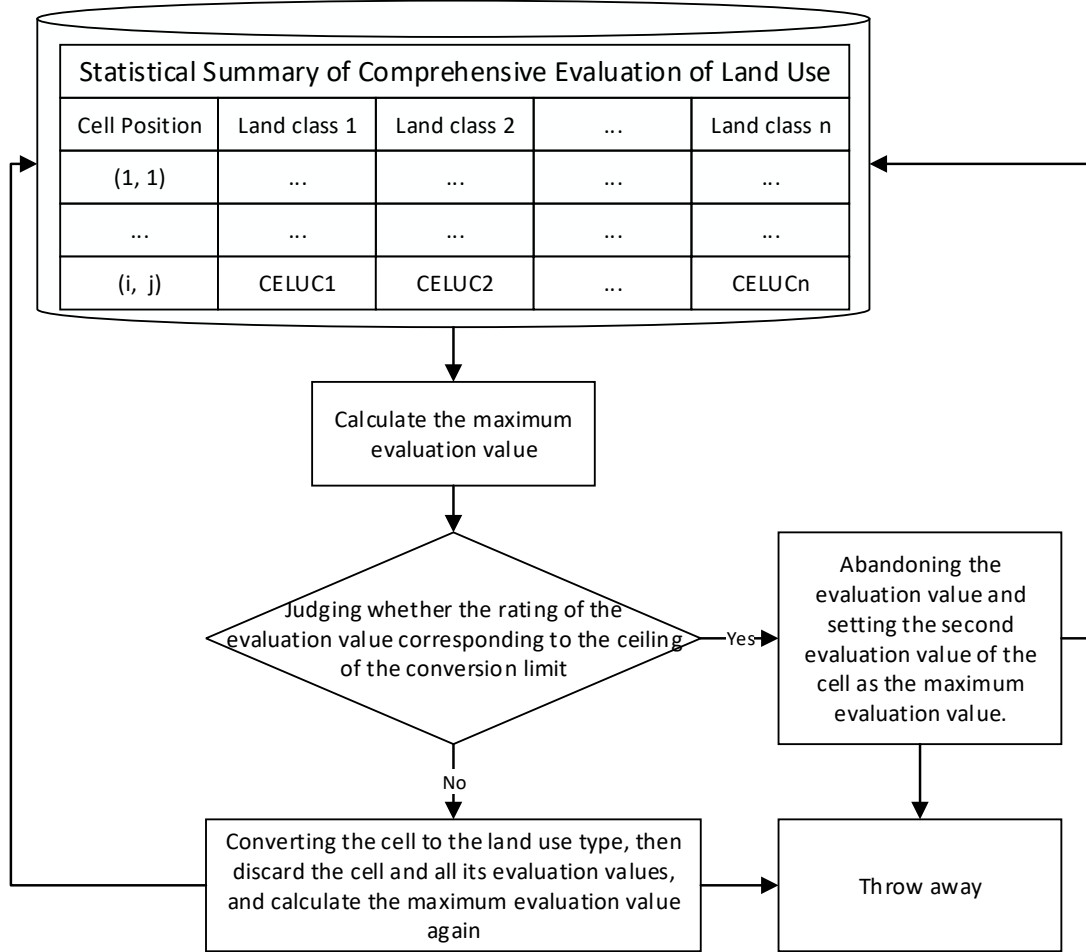

**Figure 9.** Cell's land-use-type conversion flow.

If the upper area limit is not reached, the land-use type of the cell would be converted into a new land-use type and stored as a key-value pair $((i, j), \text{CELUC}i)$ into an array, discarding the other CELUCs to make sure the cell is not used in the subsequent steps. Then, it returns to the first step.

⑤    Repeating the above steps until all cells completed conversion, and finally obtaining the prediction of the whole land-use-change distribution, which was stored as an array. Each item of the array was a key-value pair $((i, j), \text{CELUC})$.

## 4. Results and Discussion

### 4.1. Model-Efficiency Analysis

One machine was used as the master node for the work of NameNode and JobTracker, and four other machines were used as the slave nodes for the work of DataNode and TaskTracker. The operating environment of the machines was the CentOS 7.1.1503 system with Java version 1.8.0_112 and Hadoop version 2.7.3. The experiment environment configuration is shown in Table 4. The hardware environment of the serial algorithm was the same as the hardware of the Hadoop node.

| IP Address | Node Role | CPU | RAM |
|---|---|---|---|
| 192.168.128.1 | Master/Namenode/Jobtracker | Four-core 2.4 Ghz | 4 G |
| 192.168.128.2 | Slaves/Datanode/Tasktracker | Four-core 2.4 Ghz | 4 G |
| 192.168.128.3 | Slaves/Datanode/Tasktracker | Four-core 2.4 Ghz | 4 G |
| 192.168.128.4 | Slaves/Datanode/Tasktracker | Four-core 2.4 Ghz | 4 G |
| 192.168.128.5 | Slaves/Datanode/Tasktracker | Four-core 2.4 Ghz | 4 G |

The running efficiency and acceleration ratio results of the serial-Markov algorithm relative to Cloud-Markov algorithm are shown in Figure 10. The running efficiency and acceleration ratio results of the serial-CELUC algorithm relative to Cloud-CELUC algorithm are shown in Figure 11. As shown in Figures 10a and 11a, the abscissa axis indicates the number of the cells (1 $n$ is approximately 9,000,000) and the ordinate axis indicates running time.

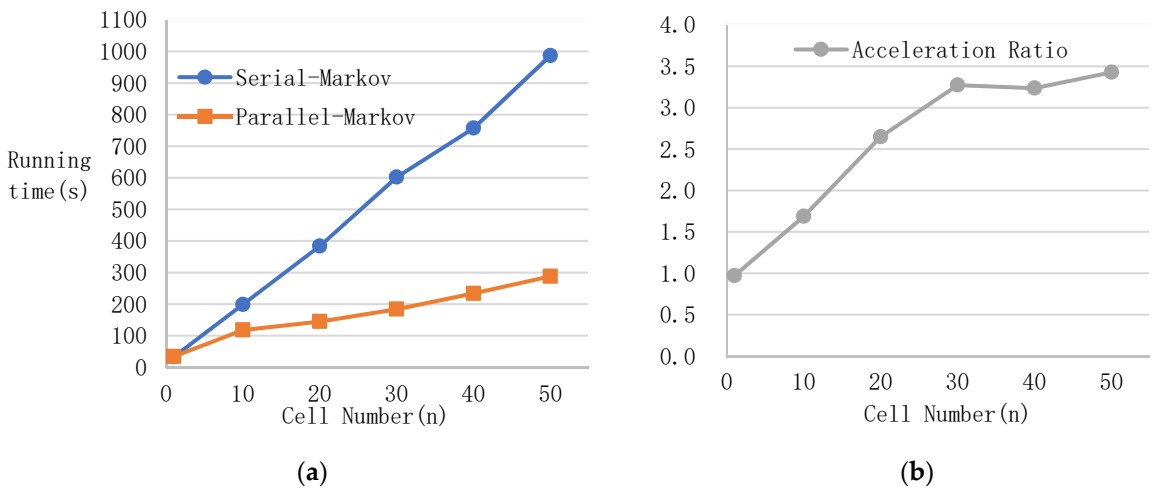

**(a)**                                     **(b)**

**Figure 10.** Running efficiency and acceleration ratio of Cloud-Markov relative to serial-Markov. (**a**) Running efficiency comparison, (**b**) acceleration ratio.

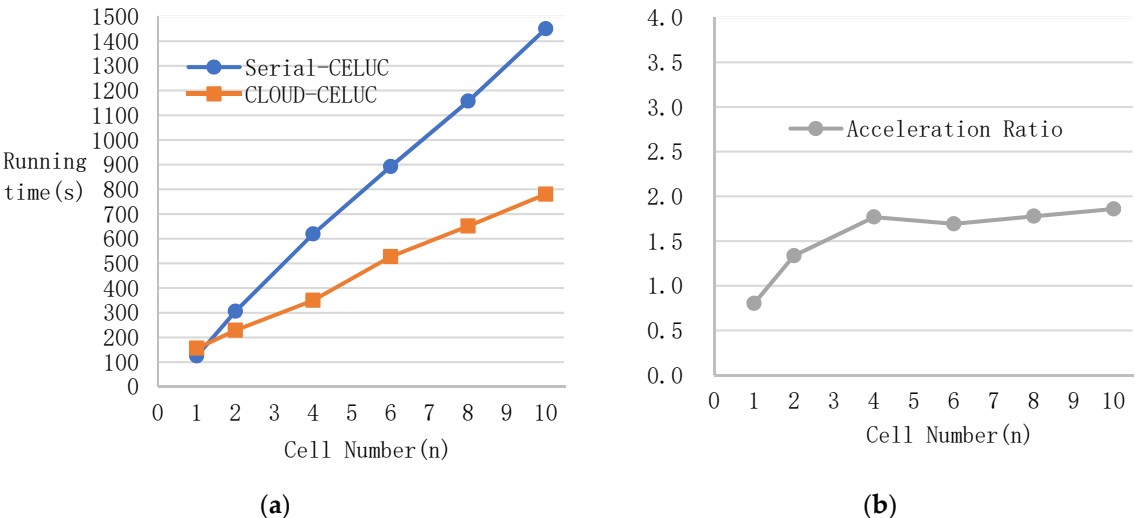

**(a)**                                     **(b)**

**Figure 11.** Running efficiency and acceleration ratio of serial-CELUC relative to Cloud-CELUC. (**a**) Running efficiency comparison, (**b**) acceleration ratio.

The results showed that the execution time of Cloud-Markov was less than that of the serial Markov algorithm, and with the increase of input data, the acceleration ratio increased and tended to

smooth. The acceleration ratio of the Cloud-Markov algorithm to the serial Markov algorithm tended to be steady at 3.27, and the acceleration ratio of Cloud-CELUC to serial CELUC tended to be steady at 1.77. The highest acceleration ratio of Cloud-Markov could reach 3.43, and the highest acceleration ratio of Cloud- CELUC could reach 1.86.

The efficiency of the parallel Markov model based on a cloud environment was remarkable because the MapReduce system effectively distributed the workload of the two phases of cell matching and quantitative statistics. The efficiency of Cloud-CELUC was also improved because the Map phase we defined ran very fast. However, when the output of the Map phase was input into the Reduce phase, it occupied a relatively large part of the long running time, which reduced the efficiency of the operation.

### 4.2. Precision Evaluation and Result Analysis

4.2.1. Precision Evaluation

The 2006 remote-sensing images and other data were defined as the initial data, and the 2013 land-use changes were simulated by suing the parallel CA-Markov model. In our experiment, the water area was fixed with no change. Based on the 2006 and 2013 land-use data, the area transition-probability matrix could be calculated. Table 5 is the 2006–2013 area transition matrix, in which each cell represents the total area of one land-use type transferring to another land-use type from 2006 to 2016. Table 6 is the 2006–2013 transition-probability matrix, in which each cell represents the probability of one land-use type transferring to another land-use type from 2006 to 2016.

**Table 5.** The 2006–2013 area transition matrix (unit: $km^2$).

| 2013 \ 2006 | Agricultural Land | Construction Land | Nature Reserve | Total |
|---|---|---|---|---|
| Agricultural land | 1282.95 | 409.71 | 95.67 | 1788.33 |
| Construction land | 210.94 | 1381.69 | 12.52 | 1605.15 |
| Nature-reserve land | 93.39 | 50.79 | 4409.60 | 4553.78 |
| Total | 1587.28 | 1842.19 | 4517.79 | 7947.26 |

**Table 6.** The 2006–2013 transition-probability matrix (unit: %).

| 2013 \ 2006 | Agricultural Land | Construction Land | Nature Reserve |
|---|---|---|---|
| Agricultural land | 71.74 | 22.91 | 5.35 |
| Construction land | 13.14 | 86.08 | 0.78 |
| Nature-reserve land | 2.05 | 1.12 | 96.83 |

As shown in Tables 5 and 6, the biggest land-use-type transition was agricultural land transferring to construction land; its ratio reached 22.91%. In order to evaluate the simulated precision, the 2013 land-use data were classified using real 2013 remote-sensing images at the data-processing stage. The simulated land-use data and the classified 2013 land-use data are shown in Figures 12 and 13, respectively.

After simulation, the precision evaluation experiments were done to correct all kinds of weight parameters defined in C-MCE. After a great number of repetitive experiments and weight-parameter corrections, the weight parameters of the suitability factors were obtained, which are listed in Table 2. At present, commonly used precision evaluation methods include visual comparison, dimensionality tests, pixel contrasts, and the Kappa coefficient test.

With visual comparison, it was found that the simulation of the nature reserve in the western and southern regions was the most accurate. It can be concluded that the suitability factor and the constrained factor were in line with the change trend of the nature reserve in the study area. The construction land

of the central urban area had better simulation accuracy. However, the construction land of east and north, including the towns of Yuanpu and Linpu, was scattered and intertwined with agricultural land. Therefore, simulation error was relatively large.

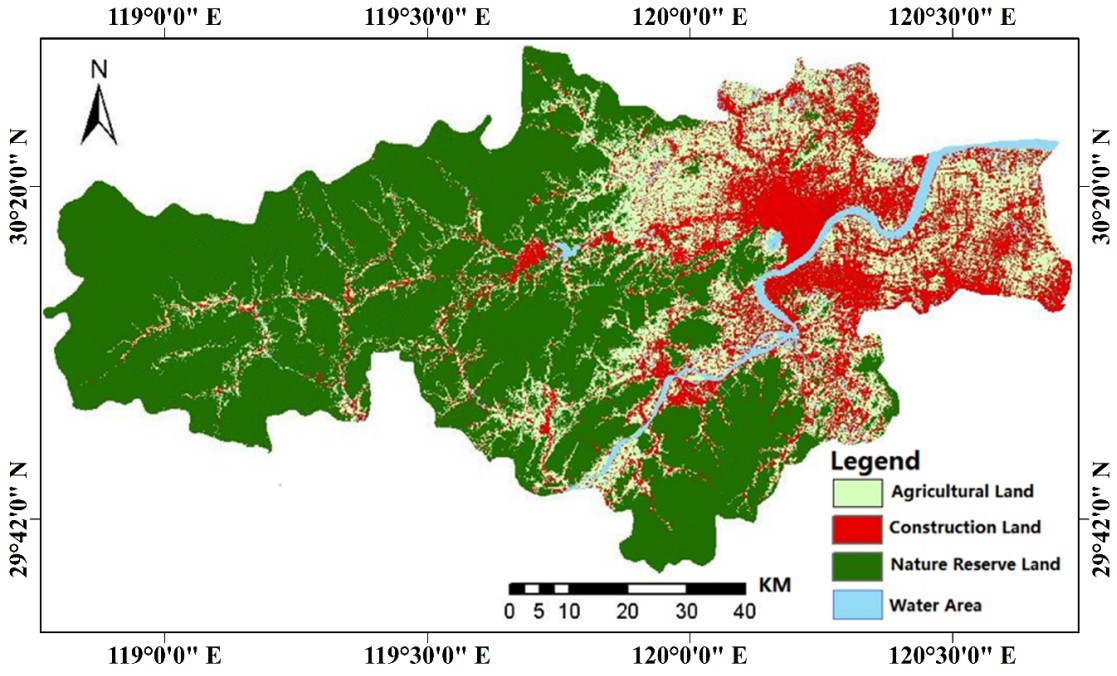

**Figure 12.** Map of land-use simulation in 2013.

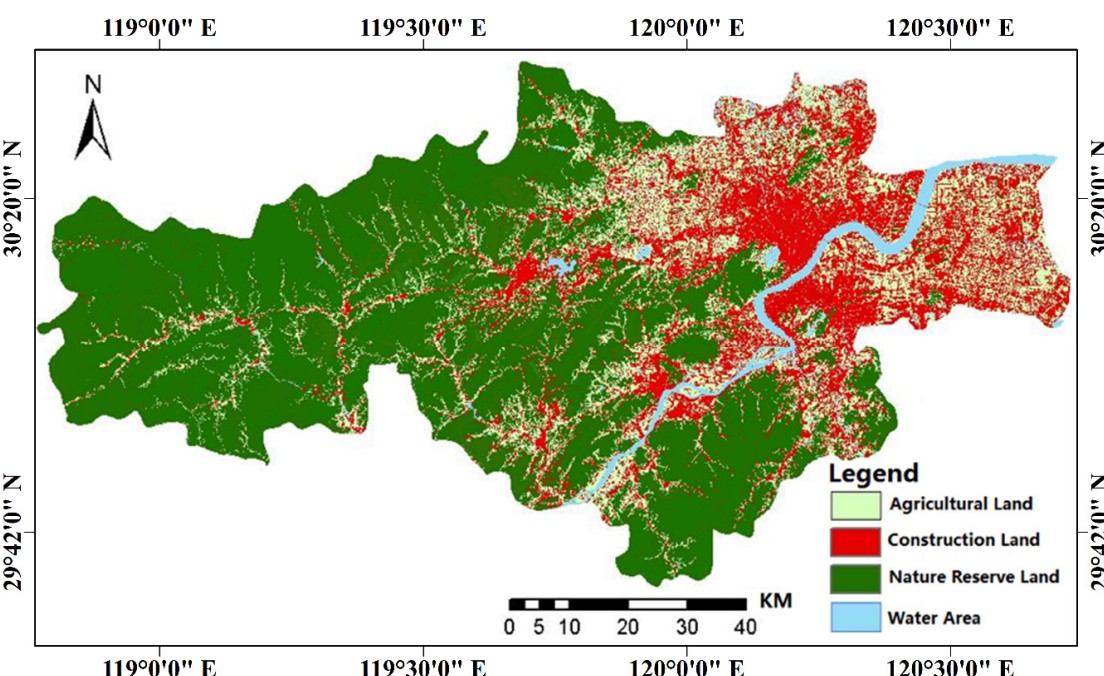

**Figure 13.** Classification map of remote-sensing images in 2013.

The Kappa coefficient test is the most commonly used quantitative test method [48]. When the Kappa coefficient is used to compare the consistency of data, commonly used criteria are as follows: If the two land-use maps are identical, then Kappa = 1; when Kappa > 0.8, consistency is almost perfect; when 0.6 < Kappa ≤ 0.8, consistency is substantial; when 0.4 < Kappa ≤ 0.6, consistency is moderate; when 0.2 < Kappa ≤ 0.4, consistency is slight; when 0 < Kappa ≤ 0.2, consistency is poor [49,50].

Simulated land-use data and actual classified land-use data of 2013 were compared, and the results of the Kappa coefficient are shown in Tables 7–9, respectively.

**Table 7.** Kappa coefficient test table of nature reserve land in 2013 (unit: km$^2$).

| Simulated Data / Classified Data | Nature Reserve | Non-Nature Reserve | Total | Accuracy | Kappa |
|---|---|---|---|---|---|
| Nature-reserve land | 4221.35 | 288.47 | 4509.82 | 93.60% | |
| Non-nature-reserve land | 296.44 | 3431.50 | 3727.94 | 92.05% | 0.86 |
| Total | 4517.79 | 3717.97 | | | |

**Table 8.** Kappa coefficient of construction land in 2013 (unit: km$^2$).

| Simulated Data / Classified Data | Construction Land | Non-Construction Land | Total | Accuracy | Kappa |
|---|---|---|---|---|---|
| Construction land | 1391.41 | 452.77 | 1844.18 | 75.45% | |
| Non-construction land | 450.78 | 5942.80 | 6393.58 | 92.95% | 0.68 |
| Total | 1842.19 | 6395.57 | | | |

**Table 9.** Kappa coefficient of agricultural land in 2013 (unit: km$^2$).

| Simulated Data / Classified Data | Agricultural Land | Non-Agricultural Land | Total | Accuracy | Kappa |
|---|---|---|---|---|---|
| Agricultural land | 1152.64 | 427.17 | 1579.81 | 72.96% | |
| Non-agricultural land | 434.65 | 6223.30 | 6657.95 | 93.47% | 0. 66 |
| Total | 1587.29 | 6650.47 | | | |

Results showed that the Kappa coefficients for nature reserve, construction land, and agricultural land were 0.85, 0.6, and 0.65, respectively. This meant that the 2013 results of the simulation were quite accurate [38,51–53], and that using the parallel CA-Markov model to predict future land use would be highly reliable.

4.2.2. Land-Use-Change Prediction

Based on the classified 2013 land-use data and other experiment data, 2020 land-use changes were predicted using the parallel CA-Markov model. The results of the 2013–2020 area transition matrix are shown in Table 10, and the 2020 land-use prediction map is shown in Figure 14.

**Table 10.** The 2013–2020 area transition matrix (unit: km$^2$).

| 2020 / 2013 | Agricultural Land | Construction Land | Nature Reserve | Total |
|---|---|---|---|---|
| Agricultural land | 1133.35 | 361.94 | 84.52 | 1579.81 |
| Construction land | 242.35 | 1587.45 | 14.38 | 1844.18 |
| Nature-reserve land | 92.49 | 50.30 | 4367.03 | 4509.82 |
| Total | 1468.19 | 1999.69 | 4465.93 | 7933.81 |

As can be seen from the 2020 land-use prediction map, construction land in the study area was on the rise as a whole, and this increase mainly came from the conversion of agricultural land. Agricultural land showed a downward trend, and natural-reserve land changed little in proportion to its vast size.

The overall growth of construction land was relatively large. In particular, due to expanding road and public-transport systems, construction land grew faster in the urban center of each county. The growth of construction land in the districts of Xihu, Gongshu, Xiacheng, Binjiang, and Xiaoshan was prominent. Due to the terrain and water-body restrictions, other counties and urban areas maintained stable acreage of construction land.

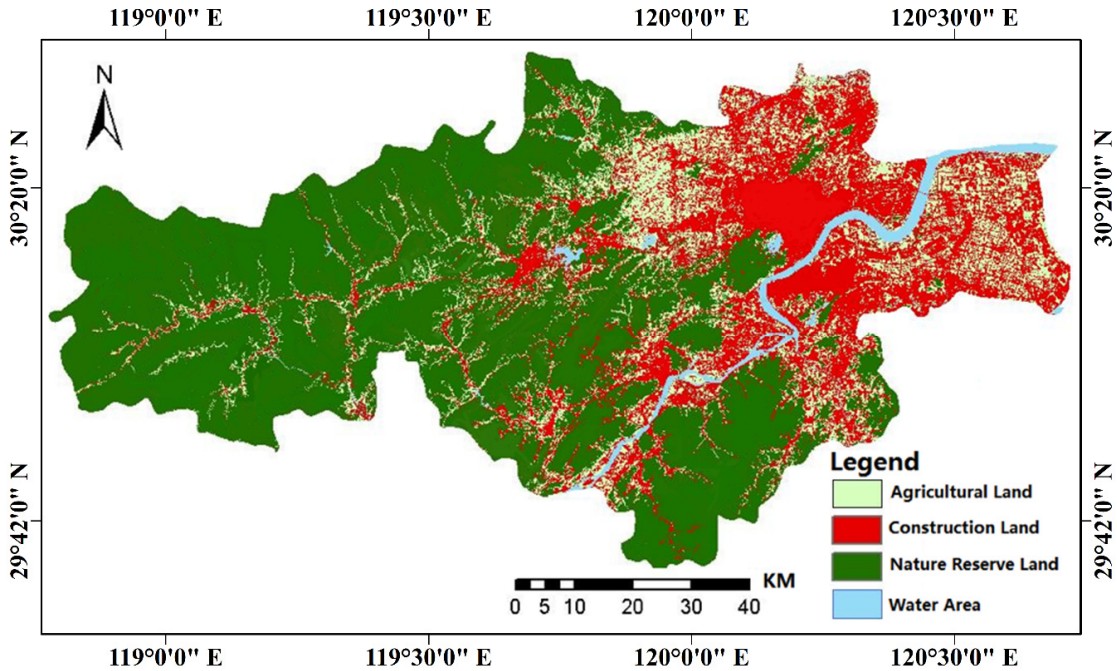

**Figure 14.** The 2020 land-use prediction map.

Agricultural land and construction land intertwined in large areas in the northwest and north of the study area. However, the evaluation value of construction land in this area was not high because it was far from the city center and the county center and the public transportation system was less developed. The acreage of agricultural land in the above area was basically stable. The agricultural land in the rest of the study area was affected by urban expansion, the road, and public transportation systems, and large acreage was converted to construction land.

Nature reserves were mostly found in the western and southern hilly areas, which were subject to various restraints for development, such as sloping terrain and ecological-protection restrictions, and an inconvenient transportation system, and therefore remained basically stable in acreage.

## 5. Conclusions

Experiments showed that the results of land-use simulation based on the CA-Markov model under a cloud environment were reliable, reflecting that the method proposed in this study was reliable and applicable. Meanwhile, MapReduce was effective in parallelizing the CA-Markov model to improve the processing speed of land-use-change prediction based on the CA-Markov model. This method parallelized the CA-Markov model in two parts: The parallel Markov model based on a cloud environment (Cloud-Markov), and the comprehensive evaluation method of land-use changes based on MapReduce (Cloud-CELUC). By selecting Hangzhou as the study area and setting up a Hadoop experiment environment, the experiments were designed to verify the reliability, precision, and operating efficiency of the method. Land-use changes in Hangzhou in 2020 were simulated and the results were analyzed. The experimental results showed that the method which simultaneously realized the integrity and segmentation for land use change simulation and prediction is also practical and effective.

This research has successfully applied the MapReduce framework to improve land-use-change prediction efficiency. However, there are still some important issues worth further investigation. First, land-use changes were restrained not only by natural conditions, but also by political, economic, demographic, and other complex factors. Due to limited data sources, this study built its model mainly on traffic, terrain, and location factors. If more data are available, the prediction module of the spatial pattern of land-use, social, economic, policy, demographic, and other factors should be taken into

account in future research. We will also consider combining the auto image-identification method [37] into our current work to reduce manual preprocessing work. When Cloud-CELUC acquired the output result of the Map stage in the Reduce stage, input/output (IO) became a bottleneck in system performance. Therefore, more research efforts should be dedicated to testing increasing IO efficiency on the performance of cloud computing based on the CA-Markov model.

**Author Contributions:** Conceptualization, J.K. and L.F.; Methodology, J.K. and S.L.; Software, J.K. and S.L.; Validation, L.F.; Formal analysis, J.K. and S.L.; Investigation, J.K.; Resources, L.F. and X.W.; Data curation, S.L. and J.K.; Writing—original draft preparation, J.K.; Writing—review and editing, L.F., J.K., and S.L.; Visualization, J.K.; Supervision, L.F. and X.W.; Project administration, J.K. and X.W.; Funding acquisition, L.F. and X.W.

**Funding:** This work was supported by the National Key Research and Development Program of China (Grant No. 2016YFC0803105, 2016YFC0502700), the China Postdoctoral Science Foundation (Grant No. 2018M641926), the China Scholarship Council Program (No. 201808360065), and the Jiangxi Provincial Department of Education Science and Technology Research Projects (Grants No. GJJ150661).

**Acknowledgments:** We would like to thank Zhejiang University GIS Lab for providing the data, and Kaibin Zhang for helping setup the experimental environment and evaluating our algorithms, and the anonymous reviewers for their valuable suggestions.

**Conflicts of Interest:** The authors declare no conflict of interest.

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
