# Peer review of "Parallel Cellular Automata Markov Model for Land Use Change Prediction over MapReduce Framework"

_ijgi, doi:10.3390/ijgi8100454_

Round 1

Reviewer 1 Report

Dear authors,

I am okay with the corrections made. Thanks.

Author Response

We appreciate the reviewer’s positive assessment and recognition of the paper.

Reviewer 2 Report

Most of my previous comments were address. However, there are some concerns remain.

In the previous version, I already mentioned the figure captions and table captions problem. They are too short and carry no explanation. The authors should write at least a longer sentence, which briefly describes the main content of the table/figure. At the end of the introduction section, the authors should add one paragraph to describe their main contributions. In section 2.2, there should be a reference for the map-reduce The authors should check all the text where the figures were mentioned, I think some of them are wrong. For example line 128, is that figure 2 or figure 3? The authors should indicate the layer index clearly in Figure 3. (where is layer 1, 2, 3, 4 in the figure). Also, the text in Figure 3, 4, 5, 8 are too small, please consider making them bigger. Please consider combining Figure 10 and Figure 11, because it can show the comparison clearer. Please consider combining Figure 12 and Figure 13, because it can show the comparison clearer.

Author Response

All the revision and edits have been highlighted in red in the manuscript for the convenience of the Editor and Reviewers.

In the previous version, I already mentioned the figure captions and table captions problem.

They are too short and carry no explanation. The authors should write at least a longer sentence, which briefly describes the main content of the table/figure.

Each figure and table are added a brief description.

At the end of the introduction section, the authors should add one paragraph to describe their main contributions.

At the end of the introduction, the main contributions have been added.

In section 2.2, there should be a reference for the map-reduce

Reference 42 was added for the map-reduce.

The authors should check all the text where the figures were mentioned, I think some of them are wrong. For example line 128, is that figure 2 or figure 3?

All the error text of the mentioned figures have been modified.

5 The authors should indicate the layer index clearly in Figure 3. (where is layer 1, 2, 3, 4 in the figure).

Figure 3 was modified to make it clearer, and a brief description was modified from line 132.

Also, the text in Figure 3, 4, 5, 8 are too small, please consider making them bigger.

The font size in Figure 3, 4, 5, 8 has been enlarged.

Please consider combining Figure 10 and Figure 11, because it can show the comparison clearer.

Figure 10 and Figure 11 have been combined.

Please consider combining Figure 12 and Figure 13, because it can show the comparison clearer.

Figure 12 and Figure 13 have been combined.

Reviewer 3 Report

This new version of the manuscript has significantly improved the quality of the material presented. The authors have addressed all the recommendations I personally suggested in the previous review round and have included more graphics and information that have completed the areas where amendments where demanded.

My congratulations to the authors.

Author Response

(The authors gave the same response as above.)

Round 2

Reviewer 2 Report

The authors solved all of my concerns. Therefore, I recommend this paper for publication.

This manuscript is a resubmission of an earlier submission. The following is a list of the peer review reports and author responses from that submission.

Round 1

Reviewer 1 Report

Thanks very much authors for this interesting piece of work.  I would like to recommend a few alterations to enhance the quality of the manuscript.  

Change (Figure 2.). It is not related with all the parameters listed and included in the manuscript and looks clumsy.

Please explicitly demonstrate the methodology through a schematic flow diagram and corresponding write up. Methodology section is not at all clear in the manuscript.

.

Please provide Geo-referenced images for figure 12, 13 and 14.

English language requires stringent revision and hence resubmission.  The manuscript is not publishable yet having the present quality.

LULC changes overtime should be demonstrated through proper statistical tables and graphs.

References have been extremely scanty and proper journal format has also not been followed.  Please review the entire literature based on CA and resubmit again.

All the data, maps, tables, graphs and flowcharts seem disconnected from each other. The manuscript has no step by step linked paragraphs and readability. The authors need to work on that and make it readable.

Author Response

We appreciate the reviewers' positive assessment of the paper and valuable suggestions on further improvements.

We thank the reviewer for the recommendation of these useful references.

1) The abbreviation in Figure 3 (previous Figure 2) has been modified and added the description in section 3.1.1, and the relation between all parameters and subsequent section has been explained here.

2) Figure 4 and section 3.1.2 have been added to demonstrate the method flow.

3) Figure 14,15 and 16 (previous Figure 12,13 and 14) has been changed to the Geo-referenced images.

4) The manuscript will be submitted to the MDPI English editing service.

5) Table 5,6,7,8,9, and10 have been modified. Table 5 was the table of 2006-2013 year area transition matrix (unit: km2), Table 6 was the table of 2006-2013 year transition probability matrix (unit: %), the land use data of 2006 was classified from 2006 remote sensing data, and the 2013 land use data was simulated by our parallel CA-Markov model.

Table 7,8,9 were the 2013-year Kappa coefficient test tables of natural reserve land, construction land, and agricultural land (unit: km2), Kappa coefficient tests were conducted by comparing 2013 simulated land use data and 2013 actual classified land use data.

6) References have been modified according to the IJGI journal format. The literature based on CA has been added in the introduction from line 32.

7) Section 3.1.1 has been added to describe the relationship between our method and future sections, and section 3.1.2 has been added to explain our method step by step.

Reviewer 2 Report

This paper designs a parallel CA-Markov model based on the MapReduce framework for land use change prediction. There are some issues that need to be addressed.

The introduction, the motivation of the paper needs to be articulated far more clearly.

In the related work section, a more rigorous investigation on the existing methods, such as comparison of previous approaches in terms of pros and cons, should be given. 

I recommend that the authors add more research related to the application of UAV for land.

Sankey, Temuulen T., et al. "UAV hyperspectral and lidar data and their fusion for arid and semi‐arid land vegetation monitoring." Remote Sensing in Ecology and Conservation 4.1 (2018): 20-33.  

Dang, L. Minh, et al. "UAV based wilt detection system via convolutional neural networks." Sustainable Computing: Informatics and Systems (2018). 

I believe that it will make this paper stronger if the authors present insightful implications in at least one paragraph based on their experimental outcomes. 

Please go through the entire manuscript to double check accuracy/consistency of each equation, table, figure and reference, and ensure English grammar error-free.  

Figure 6's caption, what is the algorithm

Line 237, the steps are wordy, please make it shorter or break it down.

Line 257, what preprocess and interpret steps were applied for the images?. Please explain this part in detail.

Please double check all the table headers, they did not follow the same rule.

Instead of showing too many figures, there should be more explanation  (For example, Figure 8, 9, 10, 11). Similarly, the tables' contents should be described more.

Author Response

We appreciate the reviewer’s positive assessment of the paper and valuable suggestions on further improvements. More efforts have been dedicated to the refinement of the paper based on the reviewer’s following comments.

1) The literature based on CA has been added in the introduction.

And we concluded that the parallel CA method cannot deal with the connection between the partitions after the research area is divided into several pieces so that the result of the segmentation will produce different final prediction results; while the traditional CA-Markov method is able to maintain the integrity of the entire study area. And for the CA-Markov model, the key problem of "segmentation and connection" remains unresolved, there was little research on the parallel CA-Markov model for land use change prediction over MapReduce framework.

2) Thanks for the recommendation, we read these two papers thoroughly.

These two papers were really beneficial for our paper modification and giving us very useful thoughts for future work. we will consider adding an auto object identification method in our current work to reduce the preprocessing work, and the structure and the thinking of these papers really helped us a lot for our modification.

3) We have checked the accuracy/consistency of each equation, table, figure and reference, and the manuscript will be submitted to the English editing service of IJGI.

4) Figure 8(previous Figure 6)’s caption changed to “Cloud-CELUC algorithm”

5) Line 237, now is Line 309 in section 3.4, we added descriptions of the input data from Cloud-Markov and Cloud-CELUC, and added a description of the output data at the last step. In each step, we added a description of how the data were used and changed.

6) Line 257, now is line 75 in section 2.1, we added Figure 1 and the related description to explain our image classification work. We used the existing software ENVI and ArcGIS to do the images classification to get the land use data, so we can focus on the land use prediction, in our future work, we will consider adding auto object identification in our work to reduce the manual work.

7) All the table headers have been checked, the unit and the date of the data was added to table 5,6,7,8,9,10 header, and added the related experimental data explanation, the experimental data of table 5, 6 was 2006 classified land use data and 2013 simulated land data. The experimental data of table 7, 8, 9 was 2013 classified land use data and 2013 simulated land use data. The experimental data of table 10 was 2013 classified land use data and 2020 simulated land use data.

The classified method was described in section 2.1, and our parallel CA-Markov model was used to obtain the simulated land use data.

8) Section 4.1 “the model efficiency” has been modified, Figure 10, 11 and 12 (previous Figure 8, 9, 10) were used to describe efficiency between the parallel algorithm and the serial algorithm, we implemented the core method of CA-Markov in the same hardware environment.

Reviewer 3 Report

The paper presents applications of CA-Markov model in land use prediction. Simulations on Hangzhou data show that the proposed method is able to reduce time cost and provide predictions. The topic is timely and interesting. However, the paper is not organized well. There are many terms used without definitions. 1) "After using overlay method to analyze two raster images with the same spatial position and 109 different times, we can obtain the transition direction of each cell, and calculate the number of cells" what does a cell represent in land use prediction? a cell in the image? 2) fig 2, 3 not clear what initial temporal data/ second temporal data represents? how the transition probability matrix is obtained? 3) section 3.2.3 Map:(?,(?,?1,?2,?3,?4,…??))→((?,?),(?1,?2,?3,…??)) ... what is cell state value? what is a uplink/downlink? 4) section 3.3 the upper area limit of 240 this land use type conversion or not It is not clear what limit represents. "If reaching the upper limit, the land use type conversion evaluation value of the cell should 242 be marked as 0, then the land use type conversion direction should be discard, and returning to 243 the first step" Authors are expected to elaborate on the rationale why cell should be marked 0 when reaching the limit. The major problem of this paper is that its contributions are not clear, i.e., what improvements/what makes it different from the previous work. In addition, based on the simulation results, it seems that the proposed method is faster but not really accurate in predictions. Authors should compare the prediction results with state-of-art methods.

Author Response

We appreciate the reviewer’s positive assessment of the paper and valuable suggestions on further improvements. More efforts have been dedicated to the refinement of the paper based on the reviewer’s following comments.

1) Each cell is a pixel in an image. We added section 2.1 to explain how vector data (we called it land use data in the paper, and there are two types land use data, one was the classified land use data which was classified at processing stage, the other was the simulated land use data which was obtained by the parallel CA-Markov model) was got, in data processing stage, we use ENVI and ArcGIS to do the image classification, then in our method, overlay method was implemented to analyze two-phase images and corresponding classified land use data, we can get the cell’s position from the images, and get the land use type from the land use data. In section 3.3, we designed a structure store this information, for example, if L1 is ‘ba-1.2234’, the evaluation value of the cell (i, j) from the initial land use type ‘b’ to the final land use type ‘a’ is ‘1.2234’.

2) Section 2.1 was added to explain how we get the classified land use data (vector data) form remote sensing images. Section 4.1 and 4.21 was modified to describe our parallel CA-Markov’s efficiency and accuracy.

In Figure 3, 5 (previous Figure 2,3), initial temporal data including the remote sensing image and the classified land use data were used to simulate the future land use change by our parallel CA-Markov model, Second temporal data were used as the reference data to analyze the accuracy of the simulated land use data.

In our experiment, firstly we classified the 2006 and 2013 year RS images to get the corresponding land use data, secondly we use the 2006 image and the 2006 classified land use data as the initial phase data to simulate the 2013 year land use change, then we tested our parallel model’s efficiency and accuracy by comparing the 2013 classified land use data and the 2013 simulated land use data, finally we used 2013 classified land use data simulate the 2020 land use change.

3) State value of the cell was calculated by using Multi-Criteria Evaluation (MCE) method which included constraint evaluation value and suitability evaluation value, the design of constraint factors and suitability factors were explained in section 3.3.2.

As shown in Figure 4, cell Kx meant that the position of the cell was in line K and row x, and we use Moore 3×3 method to calculate the neighbors impacts, so the uplink cells were K-1x-1, K-1x, and K-1x+1, the downlink cells were K+1x-1, K+1x, and K+1x+1, and as shown in Figure 5, the array structure was used to store the data in the HDFS.

4) Both the table of land use comprehensive evaluation values and the area transfer matrix of land use change were used to decide the cell’s conversion direction, where land use comprehensive evaluation values were calculated from the Cloud-CELUC stage, and the area transfer matrix of land use change was calculated from Cloud-Markov stage.

If we only use biggest conversion possibility value from the table of land use comprehensive evaluation values, it would lead to a proliferation of dominant land use types, and caused over simulations of dominant land-use types and inadequate simulation of weak land-use types. So we combined the area transfer matrix of land use change to avoid it.

"If reaching the upper limit, the land use type conversion evaluation value of the cell should be marked as 0, then the land use type conversion direction should be discard, and returning to the first step" has been modified to “If reaching the upper area limit, the CELUCi of the cell should be marked as 0, it meant one of the cell (i, j)’s CELUCi was deleted, then returning to the first step.”, for example, cell (i,j) has N kinds of land use conversion type CELUCs, when current land use type conversion was CELUCi,  after adding the cell’s area, if the total amount of the area reached the upper area limit,  then the value of the CELUCi should be marked as 0, value 0 of the CELUCi meant follow-up steps would skipped it.

5) In section 4.2.1, the accuracy experiment was conducted by using the 2013 classified land use data by actual remote sensing image, and the 2013 simulated land use data by our parallel CA-Markov model, the Kappa coefficients for the natural reserve, the construction land, and the agricultural land are 0.85, 0.6 and 0.65 respectively. It meant that the results of the simulation for 2013 was pretty accurate, and it implied that using the parallel CA-Markov model to predict future land use would be highly reliable.

Reviewer 4 Report

The submitted paper is an interesting approach to present a new model for land use change prediction by means of Cellular Automata over MapReduce. The methodology design is appropriate, the methods conducted are well grounded and the results are significant and with scientific soundness.

The structure of the manuscript is chaotic and needs a great improvement, following the main structure of scientific papers: introduction, material and methods (including a description of the study area), results, discussion and conclusions.

The introduction, as it currently stands, is too synthetical and needs an improvement, including more information about land use change detection and land use change science.

Please, follow the citation format of the journal. “References must be numbered in order of appearance in the text (including table captions and figure legends) and listed individually at the end of the manuscript”.

I do not find that the kappa coefficients of this method can demonstrate the validity of the parallel model. I agree that 0.86 is a consistent kappa figure, but how do you justify that 0.68 and 0.66 are strong figures?

Author Response

We appreciate the reviewer’s positive assessment of the paper and valuable suggestions on further improvements.

We thank the reviewer for the recommendation of these useful references.

1) The structure of the manuscript has been modified and follow the review.

2) Section 2.1 was added to explain our image classify processing experiment, we didn’t do the object detection of the image, we focused on land use prediction method, and in the future, we will enrich our method to make it more automatic.

And we will ask the English editing service of IJGI to help modify the English grammar.

3) The format of the manuscript has been modified to meet the IJGI standards.

4) Thanks for the great question. We used the kappa coefficients to compare the consistency of the two data. One was the 2013-year land use data classified by using the actual remote sensing image at the data processing stage, and the other was the 2013-year simulated land use data obtained by the parallel CA-Markov model.

From previous research [46,47], commonly used criteria are as follows: if the two land-use maps are identical, then Kappa=1; when Kappa > 0. 8, the consistency is almost perfect; when 0.6<Kappa≤0.8, the consistency is substantial; when 0.4<Kappa≤0.6, the consistency is moderate; when 0.2<Kappa≤0.4, the consistency is slight; when0<Kappa≤0.2, the consistency is poor.

So we consider 0.86 should be an almost perfect score, 0.68 and 0.66 should be a substantial score, and the results of the 2013-year simulated land use data were pretty accurate [48–51].